

# International fitness scale (IFIS): association with motor performance in children with obesity

Mariangela Valentina Puci[1,2], Caterina Cavallo[3], Alessandro Gatti[4], Vittoria Carnevale Pellino[4,5], Daniela Lucini[6,7], Valeria Calcaterra[8,9], Gianvincenzo Zuccotti[8,10], Nicola Lovecchio[11] and Matteo Vandoni[4]

[1] Clinical Epidemiology and Medical Statistics Unit, Department of Medicine, Surgery and Pharmacy, University of Sassari, Sassari, Italy
[2] Biostatistics and Clinical Epidemiology Unit, Department of Public Health, Experimental Medicine and Forensic Science, University of Pavia, Pavia, Italy
[3] Exercise and Sports, LUNEX International University of Health, Lussemburgo, Lussemburgo
[4] Laboratory of Adapted Motor Activity (LAMA)-Department of Public Health, Experimental and Forensic Medicine, University of Pavia, Pavia, Italy
[5] Department of Industrial Engineering, University of Roma "Tor Vergata", Rome, Italy
[6] BIOMETRA Department, University of Milan, Milan, Italy
[7] Exercise Medicine Unit, Istituto Auxologico Italiano IRCCS, Milan, Italy
[8] Pediatric Department, "Vittore Buzzi" Children's Hospital, Milan, Italy
[9] Department of Internal Medicine, University of Pavia, Pavia, Italy
[10] Department of Biomedical and Clinical Science, University of Milan, Milan, Italy
[11] Department of Human and Social Science, University of Bergamo, Bergamo, Italy

Corresponding author
Vittoria Carnevale Pellino,
vittoria.carnevalepellino@unipv.it

## ABSTRACT

**Background:** Overweight and obesity are defined as abnormal or excessive fat accumulation that presents a risk to health; and compared with their normal-weight peers, these individuals tend to have a lower level of self-confidence, and consequently lower physical activity adherence. Due to these self-perceived barriers, the aim of our study was to evaluate the efficacy of an online training program on self-reported physical fitness (SRPF) in children with obesity (OB).

**Methods:** A total of 32 children with OB carried out physical fitness (PF) tests and were asked to complete the International Fitness Enjoyment Scale (IFIS) questionnaire. The physical fitness tests were the Standing Broad Jump (SBJ), the 6-Min Walking Test (6MWT) and the 4 × 10 m sprint test. Children participated in a 3-weekly 60-min training session through Zoom platform. Before the beginning of the training protocol, OB children were compared with normal weight (NW) ones for PF batteries and the IFIS questionnaire. Changes in performances after the training were assessed by paired Student t and Wilcoxon tests.

**Results:** After the online training program children increased their performance in 6MWT (mean difference (MD) = 54.93; $p < 0.0001$) in SBJ (MD = 10.00; $p = 0.0001$) and in 4 × 10 m sprint test (MD = −0.78; $p < 0.0001$). No differences were found in children's physical fitness perception.

**Discussion:** Our study highlighted how a structured online training program can lead to improvements in PF of children with OB. Instead, the lack of differences in SRPF after the training suggests interesting questions to be explored on the aspects linked to self-perception. Therefore, even if our training protocol could not directly improve SRPF in children with obesity, the enhancement of their PF could be a

starting point for achieving this result with a longer training period and consequently improve PA participation for children with OB.

## INTRODUCTION

Overweight and obesity (OB) are defined as abnormal or excessive fat accumulation that presents a risk for health (*World Health Organization, 2021*) and their prevalence continues to increase, representing a serious public health concern. In children, it is known that OB has a significant impact on both physical and psychological health; in fact, childhood OB increases the risk of being obese during adulthood and it is associated with the development of comorbidities such as diabetes, cardiovascular, and other non-communicable diseases (*Sahoo et al., 2015*). The *World Health Organization (2021)* reported a worldwide increase in childhood OB over the years with values ranging from 4% in 1975 to 18% in 2016. In Italy (*Lauria et al., 2019*), data from the National population-based surveillance system showed a decrease in the prevalence of OB in primary school children, however, nevertheless, its prevalence remains among the highest in Europe (*Lauria et al., 2019*).

An unhealthy lifestyle, such as an increased consumption of ultra-processed food and the adoption of sedentary behavior concurred with the onset of OB from a young age. Children with OB compared with their normal-weight peers tend to have a lower level of physical activity (PA) with augmented self-perceived barriers to sports participation and PA practice (*Lovecchio & Zago, 2019*; *Vandoni et al., 2021b*). Many studies reported the difficulty for children with OB to start and maintain a regular PA practice due to a reduction of their physical fitness (*Arpesella et al., 2008*; *Valerio et al., 2018*) (PF, defined as the ability to perform an exercise with the correct development of physiological and psychological skills (*Ortega et al., 2008*)) and excessive body mass. Poor PA and PF levels cause a reduction of the perceived enjoyment during physical exercise and PA (*Giuriato et al., 2020*). Consequently, lower enjoyment in children with OB reduces the adherence to exercise programs and PA, which causes a negative spiral of disengagement in PA with low self-reported physical fitness (SRPF), less PA level, and poor health-related PF (*Craft, Pfeiffer & Pivarnik, 2003*; *Vandoni et al., 2021b*). A model proposed by *Stodden et al. (2008)* and lately confirmed by *Harter & Bukowski (2012)*, showed SRPF, defined as own's PF perception, as a precursor of enjoyment and a strong carrier for promoting adherence to exercise program reducing also the sports-abandon (*Lovecchio & Zago, 2019*; *Prochaska et al., 2003*; *Stodden et al., 2008*; *Wallhead & Ntoumanis, 2004*). Moreover, *Ortega et al. (2008)* demonstrated that a lower SRPF was positively related to a worst cardiovascular profile and to an increased risk of weight gain. To reduce these barriers (*Lovecchio & Zago, 2019*; *Vandoni et al., 2021b*), several authors underlined the importance of enjoyment during PA participation to increase intrinsic motivation in children to begin and maintain

long-term adherence to PA practice (*Giuriato et al., 2020*). For example, the methodological approach or the evaluation of different outcomes could be the solution (*Lovecchio & Zago, 2019*). In light of this, sport specialists and trainers should ameliorate and tailor their exercise programs based on the outcomes with better self-confidence to reduce drop-out, especially in children with OB (*Lovecchio & Zago, 2019*).

Since the COVID-19 outbreak, several countries have imposed restrictions including the closure of sports centers, therefore many aspects of everyday routine have changed, and this influenced both physical and mental health in different populations (*Carnevale Pellino et al., 2022*; *Gatti et al., 2022*; *Hallal et al., 2012*; *Robinson et al., 2021*; *Tornaghi et al., 2021*). Several studies showed a reduction in the PA level and increased unhealthy dietary habits, which increase fat mass accumulation and the risk to develop OB. Moreover, consequently to the reduction of the PA level, some studies highlighted a misperceived SRPF in children and adolescents compared to their actual level of PF and PA (*Gatti et al., 2022*; *Makizako et al., 2021*). Furthermore, since a previous study by *Vandoni et al. (2021b)* showed that the SRPF in children with OB is lower compared to their normal-weight peers, this unique situation could strongly limit participation in an exercise program for children with OB and worsen their health status.

To counter the effect of COVID-19 restrictions, trainers started to engage children with OB PA through tele-exercise (*Calcaterra et al., 2021b*; *Chen, 2018*; *Gil-Cosano et al., 2020*; *Vandoni et al., 2021a*; *Vandoni et al., 2022*); this solution allowed people to continue, even if with less training equipment, their training routines and maintain an active lifestyle. In fact, many online technologies and electronic devices were developed through the years, and, thanks to this, training programs were enhanced through web channels, applications, and online platforms (*Vandoni et al., 2021a*). The adoption of online methods to deliver training programs during training was the only solution for trainers to continue their work, although even nowadays with the reduction of COVID-19 restrictions, online training is commonly used to promote PA in many populations (*Calcaterra et al., 2021a*; *Mannarino et al., 2023*; *Sylvia et al., 2023*). This training modality allows the participant to train without seeing each other and, since children with OB tend to have more difficulties practicing group PA (*Gasser-Haas, Sticca & Wustmann Seiler, 2020*), could be a strong instrument to improve SRPF and, consequently, enjoyment and time spent doing PA (*Stodden et al., 2008*). Unfortunately, to the best of our knowledge, there is a lack of data about the effect of an online training program in children with OB PF and SRPF. For these reasons, the aim of our study is to evaluate the efficacy of an online training program on PF and SRPF in children with OB.

## MATERIALS AND METHODS

### Participants

A cohort of 32 Caucasian children with OB (BMI z-score ≥ 2, according to the World Health Organization classification) were consecutively enrolled at Vittore Buzzi Children's Hospital, Milan, Italy, from March 2021 to December 2021. Children were asked to participate in the study during a pediatric specialist visit. Children who participated in the study were aged between 8 and 13 years, had a BMI z-score ≥2, and had knowledge of the

Italian language. Exclusion criteria were known secondary obesity conditions, cardiovascular and respiratory diseases, comorbidities, orthopedic injuries, and absolute contraindications to PA practice. During the visit, children carried out physical tests to assess PF and were asked to complete the International Fitness Enjoyment Scale (IFIS) questionnaire. Both PF tests and IFIS questionnaires were conducted by two previously instructed operators always in the same place. During the execution of the IFIS questionnaire parents or guardians could assist without influencing the child. Parents or guardians gave written informed consent to participate in the study after explaining the study protocol. Children could withdraw from the study at any time with no consequences. Before the beginning of the training protocol, normal weight (NW) matched for age and gender were voluntary enrolled. Children carried out physical tests to assess PF and were asked to complete the IFIS questionnaire with the same procedure of the children with OB. The study protocol was approved by the Ethics Committee of Milan Area 1 (protocol number 2020/ST/298) and conducted in accordance with the Helsinki Declaration (*JAVA, 2013*).

## Anthropometric characteristics

All the anthropometric measures were taken during a pediatric specialist examination. Weight was measured by standing in lightweight clothing in the center of a scale (Seca, Hamburg, Germany) with hands at the sides and looking straight ahead and facing the recorder. Standing height was assessed using a Harpenden stadiometer (Holtain Ltd., Cross-well, Crymych, UK) with a fixed vertical table and an adaptable head (*Calcaterra et al., 2017*, *2018*). Then Body Mass Index (BMI) was computed by dividing the weight (kilograms) by the height$^2$ (meters squared) and transformed into BMI z-score using World Health Organization reference values (*Mei & Grummer-Strawn, 2007*).

## International physical fitness (IFIS) questionnaire

The IFIS questionnaire is a valid and reliable tool (moderate to good reliability with weighted mean Kappa: 0.70 and 0.59) that evaluates the SRPF in school-aged children (*De Meester et al., 2016*; *Ortega et al., 2011b*) retrieved by the HELENA study website (www.helenastudy.com/IFIS). The IFIS consists of a five-point Likert scale (from one very poor to five very good) with questions focused on five macro-areas of fitness: general fitness, cardiorespiratory, strength, speed-agility, and flexibility. Then we classified PF score into three different levels, as follow: 1–2 low; 3 medium; and 4–5 high perception.

## Physical fitness tests

Data collection consisted of a series of PF tests (*Skowronski et al., 2009*; *Council of Europe. Committee for the Development of Sport, 1983*). These field tests are valid and reliable tools for measuring PF in children and are widespread, inexpensive as equipment, and easy to administer (*Ruiz et al., 2011*; *Tomkinson & Olds, 2008*). All the tests were carried out by two previously instructed operators.

### Standing broad jump (SBJ)

The SBJ is a reliable and valid method (ICC ranged from 0.94 to 0.95 (*Fernandez-Santos et al., 2015*)) to evaluate the lower limb strength and power (*Carnevale Pellino et al., 2020*; *Fernandez-Santos et al., 2015*). Before the evaluation, the trainers first explained the test procedure and later demonstrated how to execute the test. Each participant started in a standing position by positioning both feet behind the starting line. After the preparatory movements, a horizontal jump was performed with the involvement of the upper limbs in free swing. Distance (to the nearest 0.5 cm) from the starting line to the heel of the back foot was registered. The test was performed twice, with a 5-min rest between each attempt, and the highest score was considered for the analysis.

### 6-minute walking test (6′MWT)

The 6MWT evaluates cardiorespiratory fitness and was conducted following the international guidelines (*Holland et al., 2014*). Children were instructed to walk the longest distance possible while maintaining their own pace. Standardized incentives and information about the remaining time, such as "You are doing well" or "Keep going" (*ATS Committee on Proficiency Standards for Clinical Pulmonary Function Laboratories, 2002*; *Vandoni et al., 2018*), were provided to the children every min. Children could stop (if required) during the test but were instructed to resume their walk once they were able to restart. The distance walked was registered in meters. A test-retest reliability analysis showed an ICC (95% CI) of 0.94 [0.89–0.96] (*Li et al., 2005*).

### 4 × 10 meters sprint test

This test is commonly used to evaluate speed-agility ability in children and requires them to sprint and turn as quickly as possible between two parallel lines 10 m apart four times. Trainers first instructed the children about how to perform the test and then showed them how to perform it. A chronograph (stopwatch W073, SEIKO, Tokyo, Japan) was used to record the time, and a shorter time indicated better performance. The 4 × 10 m sprint run test has high reliability and validity to assess speed-agility in children (*Ortega et al., 2011a*; *Vicente-Rodríguez et al., 2011*).

### Training protocol

Children with OB participated in a supervised online training program through the Zoom platform (online software, San Jose, California, USA). The exercise protocol consisted of three 60-min sessions (Monday, Wednesday, and Friday) per week for 12 weeks, for a total of 36 sessions (*Meng et al., 2022*; *Schwingshandl et al., 1999*; *Shih & Kwok, 2018*; *Son et al., 2017*). Two trainers supervised each training session and during the training, children and trainers interacted constantly. The training protocol was usually composed of 5 min of warm-up, 50 min of a combination of aerobic and strength exercises, and 5 min of cool-down or stretching. All the exercises were performed in a playful and recreative way, without the use of any equipment. An example of the training program, exercises, duration, and intensity was provided elsewhere (*Vandoni et al., 2022*).

## Statistical analysis

Quantitative variables normally distributed were summarized with mean and standard deviation (SD) whereas not normally distributed data were summarized with median and interquartile range (IQR). Qualitative variables were described by absolute frequencies and percentages. The Shapiro-Wilk test was used to assess the normality of distribution. Differences in quantitative variables between pre and post training program were evaluated using the Student t-test for paired data or its analogous non-parametric (Wilcoxon signed rank test for not normally distributed data), whereas in case of qualitative variables Pearson Chi square or Fisher Exact tests were used. Spearman's correlation coefficients were calculated to assess the relationship between the IFIS scores and motor performance parameters. For data from pre- and post-training with a $3 \times 3$ response, $p$-values were computed using the generalization of McNemar's test, commonly referred to as generalized McNemar's test or Stuart-Maxwell test for homogeneity of the marginal distributions. A $p$-value less than 0.05 was considered statistically significant. STATA 13 software was used for statistical computation.

## RESULTS

A total of 28 children with OB (19 males, mean age $11 \pm 2$ years old) completed the study protocol performing all the 36 supervised training sessions, four children with OB abandoned the training program after 12 sessions due to academic reasons. At baseline mean height was $1.52 \pm 0.11$ m, and the mean weight was $65.69 \pm 17.55$ Kg and all the anthropometrics' characteristics are shown in the Supplemental Material.

At pre-intervention assessments, OB children were compared with NW children for some anthropometric and performance measurements (Table 1).

As shown in Table 1, except for SBJ, children with NW significantly outperformed children with OB showing higher performance for 6MWT ($552.97 \pm 56.83$ vs. $479.5 \pm 61.11$ m, $p < 0.0001$) and $4 \times 10$ m tests ($14.46 \pm .06$ vs. $15.93 \pm 1.96$ s, $p = 0.001$), with similar trends for 6MWT and $4 \times 10$ m percentile values. Moreover, NW children showed significant higher median values for general physical fitness, cardiorespiratory fitness, speed agility and IFIS total score.

Table 2 shows anthropometric, motor performance, and self-reported physical fitness characteristics of the study participants after and before the training program. No differences were observed in children SRPF perception ($p > 0.05$; Table 2). Concerning motor performance, after the online training program children significantly increased their performance in SBJ (Fig. 1A), 6MWT (Fig. 1B), and $4 \times 10$ m sprint tests (in each case $p < 0.05$; Fig 1C). More in detail, in the post-training program, OB children increased their performance showing higher median values in 6 MWT ($538.5$ ($496–569.5$) vs. $471$ ($435–509$) m, $p < 0.001$) and lower median values in $4 \times 10$ m test ($15.3$ ($14.0–16.3$) vs. $15.72$ ($15.02–15.04$), $p = 0.002$). Performances improvement was also confirmed by the increase in the post-training percentiles for 6MWT ($20$ ($5–20$) vs. $5$ ($1–15$), $p < 0.001$) and $4 \times 10$ m test ($3$ ($1–10$) vs. $1$ ($1–3$), $p = 0.003$). Finally, no differences were found in children SRPF perception (Table 2).

**Table 1 Anthropometric and performance measurements at the time of pre-intervention assessments between NW and OB children.**

| Variables | NW children ($n = 29$) | OB children ($n = 28$) | $p$-value |
|---|---|---|---|
| Age (years) | 10.86 (1.7) | 10.92 (1.9) | 0.89 |
| Males (n (%)) | 17 (58.6%) | 19 (67.9%) | 0.47 |
| Height (m) | 1.48 (0.10) | 1.50 (0.11) | 0.66 |
| Weight (Kg) | 45.4 (5.6) | 65.69 (17.6) | <0.001 |
| BMI (kg/m$^2$) | 20.36 (19.56–21.22) | 27.47 (26.04–30.17) | <0.001 |
| SBJ (cm) | 103.45 (17.89) | 99.29 (20.16) | 0.41 |
| SBJ percentiles | 5 (5–10) | 5 (5–10) | 0.66 |
| 6MWT (m) | 552.97 (56.83) | 479.5 (61.11) | <0.001 |
| 6MWT percentiles | 15 (5–20) | 5 (1–15) | 0.04 |
| 4 × 10 m ST (s) | 14.46 (1.06) | 15.93 (1.96) | 0.001 |
| 4 × 10 m ST percentiles | 10 (3–30) | 1 (1–3) | <0.001 |
| General physical fitness | 4 (4–5) | 3 (2–4) | <0.001 |
| Cardiorespiratory fitness | 4 (4–5) | 3 (2–4) | 0.001 |
| Muscular strength | 4 (3–4) | 4 (3–5) | 0.41 |
| Speed agility | 5 (4–5) | 3 (2–4) | <0.001 |
| Flexibility | 4 (3–4) | 3 (2–4) | 0.39 |
| IFIS total score | 20 (19–22) | 16 (14–19) | 0.001 |

Notes:
Variables are reported as mean (SD) (data normally distributed ) and/or as median (IQR) (data not normally distributed).
NW, normal weight; OB, obesity; SBJ, standing broad jump; 6MWT , 6' minute walking test; 4 × 10 m ST, 4 × 10 meters sprint test; IFIS, international physical fitness.

**Table 2 Motor performance, and self-reported physical fitness characteristics pre and post training program ($n = 28$).**

| Variables | Pre-training | Post-training | $p$-value |
|---|---|---|---|
| SBJ (cm) | 99.29 (20.16) | 109.29 (22.83) | <0.001 |
| SBJ percentiles | 5 (5–10) | 10 (5–15) | 0.01 |
| 6MWT (m) | 471 (435–509) | 538.5 (496–569.5) | <0.001 |
| 6MWT percentiles | 5 (1–15) | 20 (5–20) | <0.001 |
| 4 × 10 m ST (s) | 15.72 (15.02–15.04) | 15.3 (14.0–16.3) | 0.002 |
| 4 × 10 m ST (s) percentiles | 1 (1–3) | 3 (1–10) | 0.003 |
| Cardiorespiratory fitness (1–5) | 3 (2–4) | 4 (3–5) | 0.05 |
| Muscular strength (1–5) | 4 (3–5) | 4 (3–5) | 0.99 |
| Speed-agility (1–5) | 3 (2–4) | 3 (3–4) | 0.36 |
| General physical fitness (1–5) | 3 (2–4) | 3 (2–4) | 0.74 |
| IFIS total score (1–5) | 16 (14–19) | 18 (15–20) | 0.35 |

Notes:
Variables are reported as mean (SD) (data normally distributed) and/or as median (IQR) (data not normally distributed).
SBJ, standing broad jump; 6MWT, 6' minute walking test; 4 × 10 m ST, 4 × 10 meters sprint test; IFIS, international physical fitness.

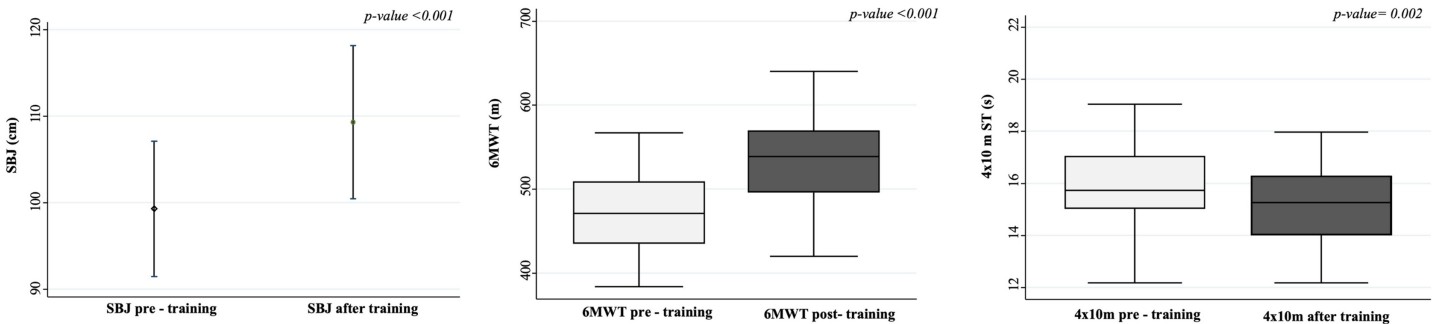

**Figure 1 SBJ, 6MWT, and 4 × 10 values pre- and after the online training.** 1A: SBJ; 1B: 6MWT and 1C: 4 × 10 m. 1A, mean values (95% Confidence interval); 1B, median (IQR); 1C, median (IQR). SBJ, standing broad jump; 6MWT, 6-minutes walking test; m, meters.

To better explore variations in SRPF due to the training program, we further classified SRPF in three different levels as follow: low-, median- and high perception. However, even stratifying by different SRPF levels, children did not change their SRPF after the training period.

# DISCUSSION

The aim of our study was to evaluate the efficacy of an online training program to improve PF and SRPF of children with OB. Our results highlighted the amelioration of children PF after 12 weeks of training, especially in muscular strength, and speed-agility PF domains. In particular, the 6MWT results reported significantly higher distances after training confirmed also by the increase of relative percentile values (from 5th to 20th percentile). In particular, the 6MWT is the most accurate test related to daily PA and cardiorespiratory fitness (*Pathare, Haskvitz & Selleck, 2012*). Our results are in line with previous studies (*He et al., 2011*; *Henriksson et al., 2020*; *Perez-Bey et al., 2020*; *Raistenskis et al., 2016*; *Tsiros et al., 2016*) demonstrating that children with OB tend to have lower levels of cardiorespiratory fitness compared to NW peers with, consequently, a higher risk of developing cardiovascular diseases in adolescence. Even if the cardiorespiratory fitness remains low, a significant improvement after 12 weeks of an online training program may become a positive starting point for children with OB to continue with the program and increase health benefits.

The SRPF plays a key role in children with OB in order to facilitate the beginning of a PA program and to assure adherence, in fact, a higher level of SRPF is related to a higher PA engagement (*Lovecchio & Zago, 2019*; *Morrison et al., 2018*; *Stodden et al., 2008*; *Vandoni et al., 2021b*). Our results suggest that children with OB may did not correctly perceive their PF abilities overestimating their performance both pre and after the training program and, although there was an improvement in PF performance after the training, the SRPF remained unchanged continuing to overestimate performances. In fact, even though children perceived their abilities as medium/good (three and four on a 5-point Likert scale), their real performances were below the 20th percentile, even after the training program. Even though, children with OB may have a misperception of their real abilities, a higher SRPF could motivate them to commit to the exercise program. In particular,

trainers and teachers should consider children preferences to tailor PA program and encourage them to maintain an active lifestyle (*Calcaterra et al., 2013*) and, in this view, the use of digital system could be an attractive way. In contrast to our results, *Goldfield et al. (2007)* showed that children with OB, who increased PA level and reduced sedentary behavior, improved SRPF due to a greater ability to understand one's own abilities and their evolution over time. Also *Morano et al. (2011)*, with a 7-month school-based intervention for children with OB, showed an improvement both in children's PF performance and SRPF. These contrasting results could be caused by the tests used in our study protocol, in fact as shown by *Southall, Okely & Steele (2004)*, children with OB find weight-bearing tasks (such as walking, running and jumping) harder to perform, as they require greater effort during movements of the body against gravity. The results of the present study confirm that children with OB have a decreased self-perception of own's physical abilities, which consequently may influence their motivation in sport participation. So, performing these types of tests might not influence the SRPF in children with OB, who already show low confidence in PF abilities compared to normal-weight peers (*Vandoni et al., 2021b*). In fact, in *Morano et al. (2014)* study, even if there was an improvement in PF perception, the results obtained in weight-bearing tasks after the training period were not correlated with PF perception. In addition, these studies used group interventions in which the presence of other peers was not beneficial for the self-perception of children with OB. Our training protocol eliminated this confrontation and, as a result, allowed children with OB to build a stronger relationship with the trainer, whom the children with OB did not perceive as if he was criticizing their performance.

Probably, longer interventions and in-person activities could be more favorable for a better perception acquisition of one's own real abilities. In fact, the possibility to directly interact with trainers helps children to better understand their abilities thanks to real-time feedback during an activity and increases their own awareness (*Frikha et al., 2019*). Also, the interaction with peers allows for a better comprehension of real abilities during the task execution (*Gasser-Haas, Sticca & Wustmann Seiler, 2020*). Although the lack of social interaction due the peculiar training context seems to be a weakness of the online program, in practice training without comparison with other children in different conditions (for example non-obese) could encourage self-confidence and therefore reduce the risk of dropout of children with OB (*Gasser-Haas, Sticca & Wustmann Seiler, 2020*). To increase the SRPF abilities and their evolution in children with OB during the execution of the online training programs, trainers or PE teachers should use frequent feedback and additional reinforcement (*Frikha et al., 2019*), as well as creating a safe environment which permits positive and constructive comparison with peers when possible (*Gasser-Haas, Sticca & Wustmann Seiler, 2020*).

This study should be interpreted in view of some limitations mainly due to the limited number of participants, therefore in the future, studies with larger sample size would be recommended to extend and validate these first results. Secondly, we did not have any control group to understand the differences between the online training program and face-to-face training in influencing children's SRPF. Finally, we are aware that some

confounding factors such as dietary patterns, socio-economical, urbanization status were not investigated in the present study.

## CONCLUSIONS

Our study highlighted how a structured training program, albeit in an online mode, can lead to improvements in the PF of children, even in peculiar categories such as children with OB. As far as SPRF is concerned, the lack of differences after the training suggests interesting questions to be explored on the several aspects linked to self-perception, primarily the interaction both with the trainer and with other children. Finally, these achievements, in addition to improving children's health status, according to Stodden and colleagues' (Stodden et al., 2008) theory in middle and late childhood, directly influence motor competence and consequently their perceived motor competence. So even if our training protocol could not directly improve SRPF in children with OB, the enhancement of their PF abilities could be a starting point to achieve this result, probably with a longer training period, and, consequently, improve PA participation for children with OB.

In conjunction with motor performance assessment, future research should evaluate other outcomes that may affect physical perception of children—regardless of obesity condition-, including social and lifestyle factors such as dietary habits, social relationship in life and school environment, and economic status.

## ACKNOWLEDGEMENTS

We would like to thank the children and their parents who generously decided to participate in this study.

### Funding

The authors received no funding for this work.

### Competing Interests

Matteo Vandoni and Vittoria Carnevale Pellino are Academic Editors for PeerJ.

### Author Contributions

- Mariangela Valentina Puci conceived and designed the experiments, analyzed the data, prepared figures and/or tables, and approved the final draft.
- Caterina Cavallo performed the experiments, prepared figures and/or tables, and approved the final draft.
- Alessandro Gatti performed the experiments, prepared figures and/or tables, and approved the final draft.
- Vittoria Carnevale Pellino performed the experiments, authored or reviewed drafts of the article, and approved the final draft.
- Daniela Lucini conceived and designed the experiments, authored or reviewed drafts of the article, and approved the final draft.

- Valeria Calcaterra conceived and designed the experiments, authored or reviewed drafts of the article, and approved the final draft.
- Gianvincenzo Zuccotti conceived and designed the experiments, prepared figures and/or tables, authored or reviewed drafts of the article, and approved the final draft.
- Nicola Lovecchio conceived and designed the experiments, performed the experiments, authored or reviewed drafts of the article, and approved the final draft.
- Matteo Vandoni conceived and designed the experiments, authored or reviewed drafts of the article, and approved the final draft.

## Human Ethics

The following information was supplied relating to ethical approvals (i.e., approving body and any reference numbers):

The Ethics Committee of Milano Area 1 approved the study (protocol number 2020/ST/298).

## Data Availability

The raw data are available in the Supplemental Files.

## Supplemental Information

Supplemental information for this article can be found online at http://dx.doi.org/10.7717/peerj.15765#supplemental-information.

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
