# Peer review of "International fitness scale (IFIS): association with motor performance in children with obesity"

_PeerJ, doi:10.7717/peerj.15765_

## Round 0.1 · original submission · Minor Revisions

In addition to the minor concerns raised by the reviewers, please clarify some issues in the statistical methods.

According to the manuscript, the Shapiro-Wilk test was used to assess the normality. However, there was no description of the results of it. Please mention which variables were normally distributed or not, and if there were any variables not normally distributed, the student-t test would not be appropriate then also clarify what kind of test was used.

Reviewer 1 ·

Basic reporting

The English language is clear and scientific.
The study is properly sourced.
The structure is respected; some adjustments should be made to the presentation. The reviewer suggests to represent individual results in a figure.

Comment:
1.1 Please provide in one figure the graphical representation of the changes per individual for SBJ (cm), 6MWT(s) and 4x10m ST (s).

Experimental design

The research question is relevant and the objective is clear.
The method is sufficiently described to be replicable.
Statistics are adequate.
Anonymized videos or photographs could be added for illustration purposes.

Comments:
2.1 Baseline is compared to normal weighted children. This should be stated in the abstract. The interventional study is not controlled. Ideally, the same NW children used for baseline should had been included for matching as control group. Could the authors discuss this important omission? Currently, only one sentence addresses this question in the limitations (l318).
2.2 Stratified analysis could be of interest: consider more advanced statistics.

Validity of the findings

The novelty is limited but the implications are interesting.
Essential data are provided.
The study is not controlled and some confounding variables are missing.
The conclusions are supported by the findings.

Comments:
3.1 Dietary patterns, socio-economical, urbanization status are not included. Were they controlled? If not available, please discuss.

3.2 line 227: "four children with OB abandoned the training program after *few* sessions due to *academic reasons*". Please provide number of sessions per individuals and detail the academic reason for withdraw.

Additional comments

Line 137: give the number of matching NW children in the text. Please clarify the deviation for Male % in Table 1, as it is a discrete variable. The text indicates 19 females (line 226), whereas the Table 1 suggests 19 males (and 67.9%). Please clarify.

Line 232: correct OB --> NB.

Line 234, 241, 245: correct 4x10 --> 4x10m

Please acknowledge the children and parents for participation.

Reviewer 2 ·

Basic reporting

.

Experimental design

.

Validity of the findings

.

Additional comments

The study is simple and solid and publishable. I would include very related papers in the discussion ( Perez-Bey A et al. Scan J Med Sci Sports 2020; 30: 1483-1496; Henriksson H et al. Eur Heart J 2020; 41: 1503-1510)

---

## Round 0.2 · Minor Revisions

Please explain the reason why original figures (Fig 1, 2, and 3) were deleted.
In addition, I could not understand which variables were normally distributed. I recommend the authors use ( ) for SDs and [ ] for IQRs to make readers clear which variables are normally distributed or not. Please unify the number of significant digits. < 0.0001 seems a little bizarre just below 0.001.

---

## Round 0.3 · Minor Revisions

It is pity that your answers for my questions in this round were out of point.

I agree that Reviewer 1 recommended you to add one figure which included the results of all tests, however, he/she did not request you to discard the original three figures. What I would like to know is the reason why did you deleted the original three figures. Please do not switch issues.

Next, I requested you to use both parentheses "()" and square brackets "[]". In medical journal articles, parentheses are often used for SD and square brackets for IQR. By using both types of brackets, we can differentiate whether variables are normally distributed or not without a lengthy explanation. Of course, this should be mentioned in footnotes.

Finally, I said, "unify the number of significant digits". It seems strange if both 0.001 and 0.0001 are presented in the same table. "< 0.001" is clearly different from "0.001" then you need not change the significant digits.

---

## Round 0.4 · Minor Revisions

Please modify "normal data" and "not normal data" before official publication. "normal data" is quite strange expression because in this case data are just "normally distributed" and the expression "normally distributed" does not mean "normal" data. First of all, what is "normal" data? Do not misunderstood the concept of the word "normal". Furthermore, the expression "not normal data" is grammatically incorrect. Please use more appropriate expression for academic journals.

---

## Round 0.5 · Minor Revisions

I recommend using a proofreading service before official publication because there are still some errors in the text.

---

## Round 0.6 · accepted · Accept

All concerns raised were now responded to appropriately.